Integration of full-length Iso-Seq, Illumina RNA-Seq, and flavor testing reveals potential differences in ripened fruits between two Passiflora edulis cultivars

Teng Yao 1 2
Wang Ye 1 2
Zhang Sunjian 2
Zhang Xiaoying 1
Li Jiayu 1
Wu Fengchan 3
Chen Caixia 1
Long Xiuqin 1
Li Anding 3 anndynlee@126.com
1 Guizhou Academy of Sciences, Guizhou Botanical Garden , Guiyang , China
2 Guizhou Academy of Sciences, Institute of Mountain Resources of Guizhou Province , Guiyang , China
3 Guizhou Academy of Sciences, Guizhou Institute of Biology , Guiyang , China
Orlov Yuriy
Electronic publication date: 2024 Sep 11
Publication date: 2024
Volume: 12
Electronic Location ID: e17983
Received 2024 Jan 25; Accepted 2024 Aug 6
Copyright: © 2024 Teng et al.
Copyright year: 2024
Copyright holder: Teng et al.
License: This is an open access article distributed under the terms of the Creative Commons Attribution License, which permits unrestricted use, distribution, reproduction and adaptation in any medium and for any purpose provided that it is properly attributed. For attribution, the original author(s), title, publication source (PeerJ) and either DOI or URL of the article must be cited.
License URL: https://creativecommons.org/licenses/by/4.0/

Keywords: P. edulis, Fruit flavor, Iso-seq, RNA-seq, Genetic difference

Funding: National Natural Science Foundation of China 31960576 National Key R&D Plan 2021YFD1100303 Guizhou Provincial Science and Technology Plan Project Qiankehe [2021] 5624 Youth Fund Project of Guizhou Academy of Sciences Qiankeyuan J zi [2023] 16 Youth Fund Project of Guizhou Botanical Garden Qi-anzhiyuan [2022] 01 Science and Technology Project of Guizhou Botanical Garden Qi-anzhikehe Z [2023] 02 Guizhou Forestry Research Project QianlinKehe [2023] 06 This work was supported by the National Natural Science Foundation of China Regional Fund Project (31960576) and the Post-subsidy project of the National Key R&D Plan (2021YFD1100303), the Guizhou Provincial Science and Technology Plan Project (Qiankehe[2021] 5624), the Youth Fund Project of Guizhou Academy of Sciences (Qiankeyuan J zi [2023] 16), the Youth Fund Project of Guizhou Botanical Garden (Qi-anzhiyuan [2022] 01), the Science and Technology Project of Guizhou Botanical Garden (Qi-anzhikehe Z [2023] 02), and the Guizhou Forestry Research Project (QianlinKehe [2023] 06). The funders had no role in study design, data collection and analysis, decision to publish, or preparation of the manuscript.

==============================
Background

Passion fruit (Passiflora edulis) is loved for its delicious flavor and nutritious juice. Although studies have delved into the cultivation and enhancement of passion fruit varieties, the underlying factors contributing to the fruit’s appealing aroma remain unclear.

Methods

This study analyzed the full-length transcriptomes of two passion fruit cultivars with different flavor profiles: “Tainong 1” (TN1), known for its superior fruit flavor, and “Guihan 1” (GH1), noted for its strong environmental resilience but lackluster taste. Utilizing PacBio Iso-Seq and Illumina RNA-Seq technologies, we discovered terpene synthase (TPS) genes implicated in fruit ripening that may help explain the flavor disparities.

Results

We generated 15,913 isoforms, with N50 lengths of 1,500 and 1,648 bp, and mean lengths of 1,319 and 1,463 bp for TN1 and GH1, respectively. Transcript and isoform lengths ranged from a maximum of 7,779 bp to a minimum of 200 and 209 bp. We identified 14,822 putative coding DNA sequences (CDSs) averaging 1,063 bp, classified 1,007 transcription factors (TFs) into 84 families. Additionally, differential expression analysis of ripening fruit from both cultivars revealed 314 upregulated and 43 downregulated unigenes in TN1 compared to GH1. The top 10 significantly enriched Gene Ontology (GO) terms for the differentially expressed genes (DEGs) indicated that TN1’s upregulated genes were primarily involved in nutrient transport, whereas GH1’s up-regulated genes were associated with resistance mechanisms. Meanwhile, 17 PeTPS genes were identified in P. edulis and 13 of them were TPS-b members. A comparative analysis when compared PeTPS with AtTPS highlighted an expansion of the PeTPS-b subfamily in P. edulis, suggesting a role in its fruit flavor profile.

Conclusion

Our findings explain that the formation of fruit flavor is attributed to the upregulation of essential genes in synthetic pathway, in particular the expansion of TPS-b subfamily involved in terpenoid synthesis. This finding will also provide a foundational genetic basis for understanding the nuanced flavor differences in this species.

Introduction

The genus Passiflora Linn., commonly known as passion fruit, is native to South America, and now cultivates in regions like Colombia, Brazil, Ecuador, Peru, as well as Southeast Asia, Australia, and New Zealand (Kugler & King, 2004; Ortiz et al., 2012). Since 1901, passion fruit has been cultivated in Taiwan Province at the turn of the 20th century and later to mainland China, over 520 species of passion fruit have now been identified worldwide (Kugler & King, 2004). Recognized for its significant economic value, passion fruit has been designated a cash crop for cultivation, especially in the southern provinces of Fujian, Guizhou, and Guangxi in China (Huo et al., 2012; Zhang et al., 2021).

Most of passion fruit are vine plants with auxiliary tendrils. Some passion fruit have been cultivated as ornamental plants owing to their beautiful flowers (Abreu et al., 2009; Santos et al., 2012). In recent years, some varieties of passion fruit become increasingly popular due to their tasty flavors and health care functions (Abreu et al., 2009; Rabanus-Wallace et al., 2021; Santos et al., 2012). Rich bioactive substances provide large numbers of nutrients that benefit human health. Passion fruit include high levels of nutrients such as vitamins, phenols, flavonoids, and minerals, and can be used as an herbal medicine to treat diseases (Qiu et al., 2020). Particularly, the fruit flavor (such as sugar-acid, vitamin C, and aroma) of cultivated passion fruit is largely loved by people. The composition of organic acids is an important component affecting the flavor of fruit juice, and its content is closely related to the quality of the fruit (Yue et al., 2015). Passion fruit belongs to the high acid type of fruits, with a suitable acid content of about 2% when consumed fresh. The addition of passion fruit juice during juice production can reduce the addition of acidulants in the juice (Gama et al., 2013; Shinohara et al., 2013). Passion fruit pulp is considered one of the very good sources of vitamin C, every 100 g of edible part contains 15–30 mg of vitamin C (Devi Ramaiya et al., 2013). In addition, the amino acid content in passion fruit is also quite rich. When comparing the fruit’s amino acid composition with the FAO/WHO reference values, leucine content has the highest value followed by phenylalanine, threonine, and valine (Shanmugam et al., 2018). As leucine plays a crucial role in growth and maintenance of the body during protein synthesis, consuming passion fruit can provide dietary nutritional levels of amino acids for the human body, and also contribute to the normal metabolism of compounds such as esters, alcohols, ketones, olefins, and aldehydes (Shanmugam et al., 2018). Besides, passion fruit has purported medicinal properties attributed to its anti-tumor, anti-anxiety (Deng et al., 2010), anti-insomnia (Deng et al., 2010), anti-inflammatory (Cazarin et al., 2016; Silva et al., 2015), antioxidant (da Silva et al., 2014; Zeraik et al., 2011), antihyperlipidemic, and antispastic efficiency (Silva et al., 2012), supporting its use as a complementary therapeutic agent. With research on passion fruit gaining momentum in China, a deeper understanding of its chemical and genetic properties could drive yield and quality improvements through breeding and cultivation practices, boosting commercial viability.

The edible varieties of passion fruit are divided into three groups, purple passion fruit (P. edulis Sims), yellow passion fruit (P. edulis Sims. f. flavicarpa Deg.) and their hybrid (P. hybrid), each featuring a range of cultivars with distinct qualities. The hybrids like “Tainong 1” (TN1) are praised for their fruit quality, whereas the purple “Guihan 1” (GH1) is known for its environmental resistance, providing excellent material for genetic studies and resource development (Xu et al., 2023; Zhu et al., 2015). Passion fruits have gained traction as an industry in Guizhou province, China, famed for its more than 100 aromatic compounds, contributing to the area’s poverty alleviation and rural revitalization efforts (Huang et al., 2022; Xu et al., 2023). Passion fruit has a short growth cycle, fast effectiveness, and good returns, which meets the needs of poverty alleviation and rural revitalization industry development in Guizhou province, China. It is becoming one of the main fruit industries in Guizhou, with a planting area of over 175,000 acres.

Plants terpene synthases (TPSs) are key rate-limiting enzymes in the formation of plant floral fragrance and fruit flavor, mainly catalyzing the synthesis of terpene compounds (Chen et al., 2011). TPS proteins have 550–850 amino acids, with a molecular weight of 50–100 KDa (Alquézar et al., 2017). Mono-TPS are located in plastids, while sesquiterpene synthase genes are located in the cytoplasm. TPS plays a crucial catalytic role in the conversion of farnesyl diphosphate (FPP) from the MVA pathway to monoterpenes and sesquiterpenes, and geranyl diphosphate (GPP) from the MEP pathway. In addition, TPS is also related to the synthesis of terpenes such as α-farnesene, linalool, and pinene (Gao et al., 2018). The TPS family can be divided into five subfamilies: TPS-a, TPS-b, TPS-c, TPS-e/f, and TPS-g. (Alquézar et al., 2017). TPS-b and TPSe/f enzymes jointly regulate the biosynthesis of floral monoterpenes in plants. TPS5 and TPS9 catalyze the production of geraniol, TPS3 is a linalool/β-ocimene synthase, and TPS4 is a linalool synthase. Cymbidium TPS18 can convert GPP into β-myrcene, geraniol, and α-pinene in vitro (Wang et al., 2021).

With the development of sequencing technologies, research on the genetic mechanisms of fruit flavor formation has gradually deepened at the gene level. In particular, the completion of the whole genome sequencing of passion fruit has greatly facilitated molecular-basis research on passion fruit flavor and precision molecular breeding. Based on the genomic sequencing and annotation of passion fruit, a total of 23,171 passion fruit genes have been identified, including 41 TPS genes (Xia et al., 2021). However, due to different objective and perspectives, the aforementioned studies mainly concentrated on the identification and annotation of genes, and did not conduct related research and verification work on the key genes regulating passion fruit flavor and their expression patterns. In order to comprehensively understand the nutritional value of passion fruit, we utilize both PacBio iso-seq and Illumina RNA-seq to sequence FL transcripts and perform a comprehensive transcriptomic analysis of two P. edulis cultivars. We focus on the purple variety (GH1) (Yao et al., 2018), with heightened environmental adaptability but decreased flavor profile, and the superior-tasting hybrid variety (TN1), (Abreu et al., 2009; Kugler & King, 2004; Santos et al., 2012). The outcomes of this study are anticipated to enhance our understanding of passion fruit’s transcriptome complexity and contribute valuable molecular insights to assist in future breeding endeavors. generating reference transcriptome sequences for passion fruit using the PacBio Iso-Seq technique and detecting transcription factors (TFs). In addition, (i) we exploring gene expression patterns and differentially expressed genes (DEGs) among the two cultivars; (ii) identifying candidate genes involved in fruit flavor. (iii) Preliminary discussion on the reasons why Passion fruit has such a pleasant smell of its fruits. This study will increase our understanding of the FL transcriptome complexity of passion fruit and provide a valuable molecular-level reference for future breeding work.

Materials and Methods

Plant materials

The passion fruit materials used from Guangxi Province for provenance tests and were planted in a nursery farm associated with the Guizhou Academy of Sciences (106.663°N, 26.714°E) in 2019. The nursery farm is situated in Qiannan Prefecture, Guizhou Province (106°48′19″E, 25°43′N), at an average altitude of 853 m, with an average annual temperature of 16.3 °C. The farm’s climate is classified as subtropical monsoon, and the soil is predominantly mountainous yellow soil. The experimental plot’s planting conditions are relatively consistent and controlled. The rainy season at the farm lasts from May to October, with the rain during this period accounting for 80% of the annual total of 1,400 mm. Both the “purple fruit cultivar” (P. edulis Sims, GH1) and the hybrid cultivar (P.edulis Sim.f flavicarpa Deg., TN1) were planted in March 2019 for germplasm conservation and provenance testing. During the fruit ripening periods, multiple tissues such as flowers, fruits, leaves, stems, and roots were harvested and immediately frozen in liquid nitrogen, then stored at −80 °C.

Measurement of biochemical indices

To evaluate the fruit quality of passion fruit, we determined four indices in ripe fruits of P. edulis and P. hybrids: total sugar (TS), total acidity (TA), vitamin C (VC), and soluble solids (SS), with at least eight individual plants (biological replicates) for each cultivar. Biochemical indices were measured using enzyme-linked immunosorbent assay (ELISA) methods according to the manufacturer’s instructions (Food Safety and Nutrition Information Technology Co., Ltd, Guizhou, China). First, the reference panels for the standards were created by mixing more solution and water in five different ratios to produce a standard curve. Then, the fruits were homogenized, and then the samples were incubated (25 °C), followed by plate washing, color development, and absorbance measurement using a microplate reader. Each assay included eight biological replicates and three technical replicates to ensure the reliability of the results.

PacBio Iso-Seq of the full-length cDNAs

Total RNAs were extracted and purified from multiple tissues, including flowers, fruits, leaves, stems, and roots. The mixed RNAs from all tissues was further sampled with equivalent solutions. The quality of the RNAs was assessed using a Nanodrop 2000, 1.0% agarose gel electrophoresis, and an Agilent 2100 (Agilent Technologies, Palo Alto, CA, USA) to ensure appropriate concentration, purity, and integrity. Only RNA with an RIN value of over 7.0 was retained. High-quality RNA was processed according to the PacBio Isoform Sequencing (PacBio Sequel II) protocol. Firstly, poly-A-tailed mRNA was isolated from total RNA using random primers with integrated Oligo (dT) magnetic beads (Clontech SMARTer™ PCR cDNA Synthesis Kit, Takara, Shiga, Japan). This was followed by first-strand cDNA synthesis with reverse transcription PCR. After optimizing the PCR cycles, large-scale PCR was employed to synthesize second-strand cDNA. Size selection was carried out with a BluePippin Size Selection System. The full-length cDNA SMRTbell libraries underwent terminal-end repair and were sequenced on the PacBio Sequel II platform. All clean data were submitted to the China National GeneBank Sequence Archive (CNSA, Project No. CNP0005167).

Illumina sequencing of the ripen fruits’ RNAs

Total RNAs were extracted from ripened fruits using the RNAprep Pure Plant Kit (Tiangen, Tianjin, China, DP441). Three biological replicates were used for library construction and Illumina sequencing to ensure the reliability. RNA quality was checked using a Qubit 2.0, 1% agarose gel electrophoresis, and an Agilent 2100 Bioanalyzer to ensure proper concentration, purity, and integrity. Subsequently, over 5 μg of total RNA was enriched using oligo (dT) magnetic beads and a fragmentation buffer. Short fragments were converted to double-stranded cDNA using random hexamers and purified with a PCR Purification Kit. After end repair and adaptor ligation, suitable-sized fragments (300 bp) were selected using 1% agarose gel electrophoresis, enriched by PCR amplification, and used to construct the cDNA library. The library was sequenced on an Illumina HiSeq 2500 platform at Frasergen, Co., Ltd. (Wuhan, China). Raw data were filtered with a Phred score >20 or sequence length >50 bp using fastp software (Chen et al., 2018); clean reads were then mapped to the referenced Iso-seq full-length sequences (NCBI, Bio Project, PRJNA418360) using bowtie2 software (Langmead & Salzberg, 2012) with default settings, except for the maximum intron size set to 5,000 base pairs. All the clean data were submitted to the China National GeneBank Sequence Archive (CNSA, Project No. CNP0005167, Submission No. sub052050). Gene expression was quantified using FPKM values as calculated by Cufflinks software (version 2.2.1) (Trapnell et al., 2012). In addition, twelve DEG were selected randomly for qPCR validation, detection methods referred to the instruction manual (Takara, Shiga, Japan). For qPCR detection, we used PeEF1 as reference gene, and then we use ΔΔCT method to obtain the results of relative expression (RQ = 2ΔΔCT).

Data preprocessing and functional annotation

PacBio Iso-Seq data were filtered with the SMRTlink suite v5. 1.0. 26412, Pacific Biosciences (http://www.pacb.com/products-and-services/analytical-sofware/smrt-analysis/). High quality subreads were achieved from the raw reads using the parameters of minimum length was 200 bp, minimum readscore was 0.65. Besides, the CCSs were obtained from the subreads by self-correction using the following parameters: minimum and maximum subread length 50 and 15,000 bp, minimum number of passes 3, minimum predicted accuracy 0.8, minimal read score 0.65, minimum accuracy of polished isoforms 0.99. The clean reads were further processed with a standard Iso-Seq3 application (https://github.com/PacificBiosciences/IsoSeq), the Circular Consensus Sequences (CCSs) were classified into Full-Length Non-Concatemer (FLNC) and non-full-length (NFL) reads according to whether or not the 5′-primer, 3′-primer, and poly(A) tail were observed. FLNC reads were performed no isoform-level clustering with the ICE algorithm to obtain FL consensus sequences (Gordon et al., 2015). The FL consensus sequences were polished by NFL reads using the Arrow algorithm to obtain high quality full length, and polished consensus isoforms. Considering higher frequency of nucleotide errors of PacBio Iso-seq reads, Illumina RNA-seq reads were further used to polishing consensus isoforms and correcting transcripts with the LoRDEC tool (Leena & Eric, 2014). Bowtie2 software (-q --sensitive --dpad 0 --score-min L,0,-0.1 -I 1 -X 1000 --no-mixed --no-discordant -p 6) were used for reads alignment. Finally, redundant sequences were removed with CD-HIT (-c 0.95, -aS 0.90) (Fu et al., 2012) to obtain the representative transcripts (unigenes) as the reference transcriptome sequences for passion fruit.

Unigenes were functionally annotated through a DIAMOND BLAST search against various public databases including NR, Swiss-Prot, KOG, Gene Ontology (GO), plantTFDB, Kyoto Encyclopedia of Genes and Genomes (KEGG), and Pfam. An e-value threshold of 1e-5 was applied during this process.

CDS and TF prediction

The protein coding sequences were predicted using TransDecoder software (https://github.com/TransDecoder/TransDecoder), the longest confident coding region was first extracted as the open reading frame, then all predicted ORFs were aligned to the Swissprot database to retain protein coding sequences as more as possible. Besides, iTAK software was used to identify the plant transcription factor (Zheng et al., 2016). iTAK is the most common tools for predicting plant transcription factors (TF), transcription regulatory factor (TR) and protein kinases (PK). The predicted proteins were submitted to online website iTAK (http://itak.feilab.net/cgi-bin/itak/index.cgi) to obtain TF/TR classification (Zheng et al., 2016).

DEG analysis and functional enrichment

Differentially expressed genes (DEGs) between the two cultivars were identified using edgeR software (Robinson, McCarthy & Smyth, 2010). Low count genes were filtered out with a CPM value of at least 2.0. The remaining genes underwent differential expression analysis with TMM normalization and were fitted to a negative binomial distribution. DEGs were selected based on a computed p-value for each gene and adjusted for multiple testing using the Benjamini-Hochberg false discovery rate (FDR ≤ 0.05). A threshold was imposed such that identified DEGs required an absolute log2-fold change of at least one in expression. Candidate DEGs were further examined for GO and KEGG enrichment using the ClusterProfiler 4.0 package (Wu et al., 2021).

Identification of candidate TPSs

BLASTP (version 2.2.3) and HMMER (version 3.0 for Windows) software (Finn, Clements & Eddy, 2011) were initially used to identify members of the TPS family. Local BLASTP searches were performed using the full-length protein sequences AtTPS1 through AtTPS23 from the Arabidopsis thaliana genome data (http://www.Arabidopsis.org/) as reference sequences. Additionally, hmmsearch was conducted in the FL iso-seq protein database using the Pfam profile for the “trehalose synthase” domain PF02358 from Pfam (http://pfam.xfam.org/). Subsequently, proteins obtained from the first step were screened to remove redundant and incomplete sequences from the N- or C-terminus using the NCBI CD-search tool. Furthermore, motif analysis was conducted to examine conserved motifs and domains using MEME software (http://meme-suite.org/tools/meme). With the manual curation, the PeTPSs and AtTPSs were obtained. For phylogenetic analysis, the protein sequences were aligned using the Mafft software (version 7) (Kazutaka, John & Kazunori, 2019), whereas conserved domain were selected using Gblock 9.1 (Castresana, 2000). The best substitution model for each locus was optimized by maximum likelihood method with 1,000 times bootstrap in the iqtree. After all the prior steps, a ML-tree was constructed with the mGTR2 distribution model, the tree file was finally output using Evoview software.

Results

Significant differences of flavors in P. edulis and P. hybrids

To assess fruit quality, four biochemical indices were analyzed: total sugars (TSs), total acidity (TA), vitamin C (VC), and soluble solids (SS) for two passion fruit cultivars. In P. edulis, the TS content was 74.97 mg/g, and TA was 25.23 mg/g, whereas P. hybrids had 64.78 mg/g TS and 19.60 mg/g TA, respectively. The VC content was 118.10 mg/g in P. edulis, compared to 59.42 mg/g in P. hybrids. The SS proportion was 12% in P. edulis and 12.25% in P. hybrids (raw data were displayed in Table S8). A two-way ANOVA revealed that SS content did not differ significantly between the ripened fruits of the two cultivars; however, P. hybrids demonstrated significantly higher levels of TS, TA, and VC than P. edulis (Fig. 1). These findings suggest that the distinct concentrations of soluble flavor substances (TS, TA, and VC) contribute greatly to the flavor differences between P. edulis and P. hybrids, with the enhanced levels in P. hybrids improving both nutritional value and taste.

Figure 1 Determination of biochemical indicators of fruit flavor.

(A) Total soluble sugar content (TSs, g·kg−1); (B) Total acid content (TA, g·kg−1); (C) Total vitamin C content (VC, g·kg−1); (D) Proportion of soluble solids (SSs, %). *p ≤ 0.05, **p ≤ 0.01, ***p ≤ 0.001, ns, not significant.

Correlation analysis between various nutritional indicators revealed a moderate relationship between TS and TA (r = 0.54, p = 0.067) and TS and VC (r = 0.64, p = 0.028). No significant correlation was found between TS and SS (r = 0.18, p = 0.57), TA and SS (r = 0.11, p = 0.73), or VC and SS (r = 0.19, p = 0.55). However, a strong linear correlation was identified between TA and VC (r = 0.94, p < 0.001) (Fig. 2). This suggests that an increase in VC in P. hybrids might lead to an elevated TA content, thereby reducing the acid-sugar ratio to 2.97 in P. hybrids compared to 3.31 in P. edulis.

Figure 2 Correlation analysis of six fruit quality indicators.

(A) Correlation between TS and TA, (B) correlation between TS and VC, (C), correlation between TS and TSs, (D) correlation between TA and VC, (E) correlation between TA and TSs, (F) correlation between VC and TSs.

SMRT sequencing

Based on PacBio Iso-Seq, a total of 28,953,014 subreads and 522,427 polymerase reads representing 34.49 Gb data were generated in passion fruit, with a mean length of 1,192 bp. The maximum and minimum length of subreads were 221,656 and 51 bp, the N50 length were 1,315 bp (Table S1). A total of 427,342 Circular Consensus Sequences were generated, with a mean length of 1,370 bp, the average accuracy was 99.79% and the average passes was over 60 cycles, the maximum and minimum length of CCS were 9,838 and 102 bp, the N50 length were 1,517 bp (Table S2). A total of 427,342 Circular Consensus Sequences (CCS) were generated, with a mean length of 1,370 bp, the average accuracy was 99.79% and the average passes was over 60 cycles, the maximum and minimum length of CCS were 9,838 and 102 bp, the N50 length were 1,517 bp (Table S2). The complete full-length transcripts contain 5′-terminal, 3′-terminal, and poly-A. When we identified FLNC sequences, 366,054 FL reads, 358,883 FLNC reads and 358,519 poly-A FLNC reads representing the full-length reads, full-length non-concatemer reads and full-length non-concatemer reads contains complete poly-A were found respectively in passion fruit (Table S3). Subsequently, we clustered the results of FLNC reads to eliminate the redundancy reads, a total of 34,457 FLNC reads were generated, with a mean length of 1,319 bp, the maximum and minimum length of FLNC were 7,779 and 200 bp, the N50 length were 1,500 bp (Table S4).

Considering the high error rate of nucleic acid bases in Pacbio Iso-Seq approach, we polished transcripts using LoRDEC software, and then removed redundant transcripts to obtain the final isoforms using cd-hit software. A total of 34,457 transcripts and 15,913 isoforms were generated, with a mean length of 1,319 and 1,463 bp, the maximum and minimum length of transcripts and isoforms were 7,779/200 bp (max/min), 7,779/209 bp, the N50 length were 1,500 and 1,648 bp respectively (Table S5 and Fig. 3A). The number of clusters with the same transcripts were clustered into 10 clusters, the most transcripts category is cluster 1 (Fig. 3B).

Figure 3 Distribution of the zero-mode waveguide (ZMW) reads and transcripts length.

(A) Read-length distribution of the zero-mode waveguide sequencing, (B) Distribution of the non-redundant transcript Length.

Illumina RNA-Seq

Illumina RNA-Seq on P. edulis and P. hybrids’ ripe fruits generate over 300 million raw reads, and the sequencing depth reaches over 135 (Table S6, each sample has three replicates, and the clean data were submitted to the China National GeneBank Sequence Archive (CNSA, Project No. CNP0005167). Post-filtering yielded massive clean reads, with each sample exceeding 45 million reads and 6.07 Gb of clean bases. Q20 and Q30 values were above 97.50% and 92.77%, respectively, and the GC content ranged from 45.86% to 46.59% (Table S6). The statistical power of RNA-seq data was calculated in RNASeqPower software (RNASeqPower 0.84, https://rodrigo-arcoverde.shinyapps.io/rnaseq_power_calc/), and the power value of each sample was over 0.9 (Table S6). Then, the clean reads were mapped to the reference isoforms using Bowtie2, with mapping rates between 75.09% to 80.51% (Table S7). To evaluate the overall expression of the transcriptome among different samples, the read count values were replaced by the fragments per kilobase of transcript per million (FPKM) values for all mapped genes. We normalized read counts to FPKM values to compare transcriptome expression across samples, filtering out genes with low expression (log2FPKM < 1) and charting the expression profiles for each sample (Fig. 4). In addition, twelve DEG were selected randomly for qPCR validation, 10 of them were consistent with the RNA seq results in expression trends (Fig. S1).

Figure 4 Expression level of Illumina RNA-seq.

Functional annotation

To annotate passion fruit unigenes derived from four different organs, a similarity search against five public databases (NR, Swiss-Prot, KOG, GO, and KEGG) was conducted (Fig. 5A). Noticeable annotation success was observed: 15,322 FL unigenes (97.61%) in NR, 2,977 FL unigenes (18.71%) in GO, 7,339 (46.12%) in KEGG, 9,677 (60.81%) in KOG, and 13,201 (82.96%) in Swiss-Prot. A total of 15,535 (97.62%) unigenes were annotated in at least one database, with 1,499 (9.42%) annotated across all databases (Table 1 and Fig. 5B), demonstrating comprehensive gene coverage in the FL transcripts of passion fruit. Homology alignment of unigenes in the NR database revealed that the top five related species included Populus trichocarpa, Hevea brasiliensis, Jatropha curcas, Manihot esculenta, and Populus euphratica (Fig. 5C), with the majority of unigenes showing high homology (E-value < 0.00001) with consistent E-value distribution across different organs (Fig. 5D).

Figure 5 Function annotion of full-length unigenes from P. edulis.

(A) Function annotion of all unigenes with five database, (B) venn plot of function annotion, (C) Species classification of NR annotion, (D) E−value Distribution in NR annotion.

Table 1 Gene annotation results from five different databases.

Database	All	NR	GO	KEGG	KOG	Swissprot	Unannotated	
Gene number	15,913	15,532	2,977	7,339	9,677	13,201	378	
Proportion	100.00%	97.61%	18.71%	46.12%	60.81%	82.96%	2.38%	

Identification of CDS and TFs from the Iso-Seq data

Coding sequence (CDS) is a sequence of DNA that encodes a protein, extending from the start to the terminating codon. We utilized TransDecoder software for CDS prediction analysis, with results presented in Figs. 6A–6D. For all isoforms, we identified 12,191 5′ UTRs with an average length of 197.61 bp. The lengths of the 5′ UTRs ranged from 3 to 2,631 bp. Most were shorter than 500 bp; only 737 (6.05%) exceeded this length (Fig. 6A). Additionally, we found 14,697 3′ UTRs averaging 287.00 bp in length and ranging from 3 to 3,508 bp, with most under 500 bp and only 970 (6.60%) over this threshold (Fig. 6B). A total of 14,822 putative CDSs were identified, with an average length of 1,063 bp and only 139 (0.94%) extending beyond 3,000 bp in length (Figs. 6C and 6D).

Figure 6 Results of CDSs and TFs annotion.

(A) The 3′UTR length distribution map, (B) the 5′UTR length distribution map, (C) the CDS length distribution map, (D) length distribution of 3′UTR, 5′UTR, and CDS, (E) the top 10 TFs in functional annotion.

In passion fruit, we identified 1,007 transcription factors (TFs) from 84 different families via the iTAK pipeline. The ten most represented TF families are displayed in Fig. 6E, including prominent families such as AP2/ERF, NAC, WRKY, C2H2, and bZIP in P. chienii. With 70 putative AP2/ERF TFs identified, this family had the largest representation, followed by NAC (53 members), WRKY (51 members), C2H2 (44 members), and bZIP (43 members). These numerous TFs provide a rich resource for further research.

DEGs and functional enrichment analysis

To identify differential gene expression between the cultivars of P. edulis and P. hybrid, we compared the paired combinations of GH1 versus TN1 to identify upregulated and downregulated unigenes in the latter relative to the former (Fig. 7A). The specifics of the DEGs in P. edulis versus P. hybrid are shown in Fig. 7B. A total of 357 DEGs were found, comprising 314 upregulated unigenes and 43 downregulated ones. Furthermore, fewer upregulated genes were observed in the TN1 cultivar compared to GH1.

Figure 7 Data analysis and enrichment of all DEGs.

(A) DEGs of the the two cultivars, (B) heatmap of all DEGs, (C) the top 10 enrichment GO terms, the left figure shows up-regulated DEG en-richment GO terms, the right figure shows down-regulated DEG enrichment GO terms.

For more detailed assessing the main biological functions and metabolic pathways, we conducted DEG enrichment analyses using GO terms and KEGG pathways. In the TN1 versus GH1 comparison, GO enrichment analysis indicated that 79 terms were significantly enriched, comprising 54 upregulated and 25 downregulated terms. Prominent categories included “oxidation-reduction process” (GO:0055114) and “extracellular region” (GO:0005576) (Fig. 7C). Upregulated genes were primarily associated with resistant functions like “response to stress” and “hydrogen peroxide catabolic process,” while downregulated genes were principally related to nutrient transport, for example, “carbohydrate transport” and “sucrose transport.” These patterns suggest that GH1 may have greater environmental adaptability, whereas TN1 may contain a richer nutrient content, as also reflected in the GO enrichment results for the fruit.

Similarly, significant KEGG pathway enrichments in the TN1 versus GH1 group were analyzed (Fig. 8). The top 20 pathways for both upregulated and downregulated categories were listed, with upregulated genes primarily involved in functions related to resistance, such as the “plant MAPK signaling pathway” and “plant-pathogen interaction”, and downregulated genes associated mainly with anabolic processes like “plant hormone signal transduction” and “glutathione metabolism”.

Figure 8 The top 20 enrichment KEGG terms.

The left figure shows up-regulated DEG enrichment KEGG pathways, the right figure shows down-regulated DEG enrichment KEGG pathways.

PeTPS identificant and phylogenetic analysis

Plants synthesize terpenes for secondary metabolites and environmental adaptation. We identified 17 TPS unigenes in passion fruit, consisting of the TPS-a subfamily (four unigenes) and the TPS-b subfamily (13 unigenes). Compared to A. thaliana TPS, there is a significant expansion in the TPS-b subfamily within passion fruit (Fig. 9). TPS genes in passion fruit are known to enhance resistance against insects and pathogens (Byun-McKay et al., 2006), while most studies focus on their role in terpenoid and aromatic substance synthesis (Hansen et al., 2017; Nomani et al., 2019). The alignment of 17 TPS protein-coding sequences from the passion fruit transcriptome showed more than 72.4% sequence similarity to the TPSs from passion fruit, and the highest sequence similarity was 86.18% (Fig. 9A). A phylogenetic tree analysis revealed that these TPS sequences fell into five clades—TPS-a to TPS-g. All passion fruit TPS sequences clustered in the TPS-a and TPS-b clades, with the TPS-a clade containing four and the TPS-b subfamily comprising 13 TPS sequences, indicating evolutionary expansion of the TPS-b family in passion fruit (Fig. 9). We also examined the expression levels of TPS genes in TN1 and GH1; 47.06% were upregulated and 41.18% were downregulated in TN1, with 61.54% of TPS-b genes upregulated (Fig. 10). This supports the hypothesis that the TPS-b subfamily may influence the fruit flavor of passion fruit. Notably, eight homologous genes at two TPS loci on chromosome 12, all belonging to the TPS-b subfamily, were detected (Xia et al., 2021). This finding, consistent with prior research, suggests significant expansion of the TPS-b subfamily, which is likely directly linked to flavor formation in passion fruit.

Figure 9 Phylogenetic analysis of AtTPSs and PeTPSs.

The five-pointed star with green color represents the AtTPSs, and the circle with yellow color represents the PeTPSs.

Figure 10 Motif and expression analysis of P. edulis TPS.

(A) Motif analysis of TPS in MEME software, (B) expression profiles of 17 TPS genes, (C–E) qPCR validation of PCL_3114, PCL_3201 and PCL_4857.

Discussion

Passion fruit is one of the novel fruit which is widely cultivated in tropical and subtropical areas (Xia et al., 2021). As an edible with medicinal function fruit plant, passion fruit contains various biologically active substances (Ma et al., 2021; Pontes, Marques & Câmara, 2009). The diverse germplasm resources of passion fruit exhibit a range of phenotypic traits; for example, the TN1 cultivar demonstrates excellent fruit quality, while the purple passion fruit GH1 cultivar is known for its cold and disease resistance. Compared to purple fruit varieties such as P. edulis Sims (GH1), P. hybrids (TN1) have higher contents of taste and flavor substances like total sugar (TS), total acidity (TA), and vitamin C (VC), enhancing their nutritional value. Interestingly, there is a moderate correlation between TS content and TA and VC contents, suggesting that an increase in VC content in P. hybrids might lead to higher TA content, consequently lowering the acid-sugar ratio in TN1 compared to GH1.

With the advance of sequencing technology, single-molecule real-time (SMRT) sequencing from PacBio provides new insights into full-length (FL) sequences, especially for non-model organisms in which it lacks reference genome sequences or those with poor-quality reference genomes. Through comprehensive analyses with PacBio Iso-Seq and Illumina RNA-Seq transcriptomic data, we obtained an overview of the gene expression profiles in P. edulis. This approach leveraged the strong complementarity of these two types of data. A previous study of the passion fruit transcriptome utilizing next-generation sequencing (NGS) technology (Illumina HiSeq2500 sequencing platform) yielded 78,192 unigenes, with average lengths of 622.78 bp and a mean N50 of 1,225 bp (Li et al., 2021; Qiu et al., 2020; Wang et al., 2023; Yi et al., 2023). In addition, the Beltsville Agricultural Research Center released a low-quality reference genome of the CGPA1 cultivar in 2017 using Illumina GAIIx sequencing technology (Santos et al., 2014). Researchers from Xiamen University (China) have completed the assembly of the chromosome level reference genome of passion fruit in 2022 (https://db.cngb.org/search/project/CNP0001287/). However, there is still a lack of shared understanding in the identification of transcripts of passion fruit. Meanwhile, the Haikou Experimental Station (Key Laboratory for Genetic Improvement of Bananas of CAS) enhanced the reference genome quality of passion fruit (Xia et al., 2021). While for revealing complete transcript information, it is still not enough. In this study, combining SMRT with Illumina sequencing technology, we generated a total of 15,913 isoforms with a mean length between 1,319 and 1,463 bp. Over 97% of unigenes were annotated in five databases, providing a comprehensive transcript reference for passion fruit flavor study and contributing to the improved annotation of the reference genomes.

FL non-redundant sequences offer extensive genetic information for transcriptional and post-transcriptional regulation, including transcription factors (TFs), long non-coding RNAs (lncRNAs), and alternative splicing (AS) events. TFs play a significant role in gene expression regulation during fruit ripening (Fan et al., 2018; Li, Chen & Grierson, 2019; Osorio, Scossa & Fernie, 2013; Wang et al., 2019). Our research identified 1,007 TFs from 84 different families; the most represented were AP2/ERF, NAC, WRKY, C2H2, and bZIP in P. chienii. We discovered 70 potential AP2/ERF TFs, more than any other TF family, followed by 53 NACs, 51 WRKYs, 44 C2H2s, and 43 bZIPs. Identifying these numerous TFs will be instrumental in supplying ample candidates for future genomics studies.

There are notable flavor differences between the TN1 and GH1 cultivars, yet the metabolic and synthetic pathways of these substances remain unclear. Our genomic, transcriptomic, and metabolomic data provide new insight into the unique biosynthetic processes of passion fruit. In the TN1 vs. GH1 comparison group, the majority of the GH1 categories pertained to essential biological functions necessary for cellular life, mainly related to stress resistance, such as the response to stress, desiccation, and the metabolic and catabolic processes of hydrogen peroxide. In contrast, the majority of TN1 genes were primarily enriched in nutrient transport functions, such as carbohydrate, disaccharide, sucrose, and oligosaccharide transport, and sucrose transmembrane transporter activity. These findings suggest that the GH1 cultivar may possess greater environmental adaptability, while the TN1 cultivar contains richer nutrient content, all these are consistent with the results of GO enrichment analysis.

Conclusions

In summary, we conducted PacBio Iso-Seq and Illumina RNA-Seq analysis on two passion fruit varieties with significant differences in fruit flavors. Interestingly, the top 10 GO and KEGG terms of TN1 (variety with better flavors) upregulated DEGs enriched in nutrient transport functions. These results of the two cultivars corresponded with the results of their performance. Furthermore, the TPS families in passion fruit have been identified in this study. Compared to A. thaliana TPS, there is a significant expansion of the passion fruit TPS-b subfamily, which possibly participates in the regulatory mechanisms of fruit flavor in passion fruit. Our findings explain that the formation of fruit flavor is attributed to the upregulation of essential genes in synthetic pathway, in particular the expansion of TPS-b subfamily involved in terpenoid synthesis. This finding will also provide a foundational genetic basis for understanding the nuanced flavor differences in this species.

Supplemental Information

Supplemental Information 1 Results of qPCR validation of RNA-seq data.

(a) ~ (i) were PCL_3727, PCL_5183, PCL_8874, PCL_16124, PCL_18357, PCL_19350PCL_18502, PCL_19539, PCL_20808, PCL_27524, PCL_30866, PCL_33444, respectively.

Supplemental Information 2 Data statistics of all subreads.

Supplemental Information 3 Data statistics of CCS sequences.

Supplemental Information 4 Results of FLNC identification.

Supplemental Information 5 Data statistics of FLNC sequences.

Supplemental Information 6 Data statistics of final transcripts and isoforms.

Supplemental Information 7 Statistics of Illumina RNA-Seq Data.

Supplemental Information 8 Mapping Stat of Illumina RNA-Seq Data.

Supplemental Information 9 Raw data for the content of influencing indicators.

Additional Information and Declarations

Competing Interests

Author Contributions

DNA Deposition

Data Availability

The authors declare that they have no competing interests.

Yao Teng performed the experiments, analyzed the data, authored or reviewed drafts of the article, and approved the final draft.

Ye Wang performed the experiments, prepared figures and/or tables, and approved the final draft.

Sunjian Zhang analyzed the data, prepared figures and/or tables, and approved the final draft.

Xiaoying Zhang analyzed the data, prepared figures and/or tables, and approved the final draft.

Jiayu Li analyzed the data, prepared figures and/or tables, and approved the final draft.

Fengchan Wu performed the experiments, authored or reviewed drafts of the article, and approved the final draft.

Caixia Chen performed the experiments, authored or reviewed drafts of the article, and approved the final draft.

Xiuqin Long conceived and designed the experiments, authored or reviewed drafts of the article, and approved the final draft.

Anding Li conceived and designed the experiments, authored or reviewed drafts of the article, and approved the final draft.

The following information was supplied regarding the deposition of DNA sequences:

The RNA-seq clean data are available at the China National Gene Bank Sequence Archive: CNP0005167.

https://db.cngb.org/search/project/CNP0005167/

The following information was supplied regarding data availability:

The raw data are available in the Supplemental Files.

The RNA-seq clean data are available at the China National Gene Bank Sequence Archive: CNP0005167.

https://db.cngb.org/search/project/CNP0005167/

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
