# Peer review of "Integration of full-length Iso-Seq, Illumina RNA-Seq, and flavor testing reveals potential differences in ripened fruits between two Passiflora edulis cultivars"

_PeerJ, doi:10.7717/peerj.17983_

## Round 0.1 · original submission · Major Revisions

This work highlights novel application area for sequencing in passion fruit genome. However the manuscript got critical remarks demanding major revision. Please check the comments by reviewer #3. Add experimental validation data (PCR). Refer to high-quality reference genomes assembled in passion fruit.

·

Basic reporting

Please revise this MS according to the following requirements:
1. There are too many descriptions in the SMRT sequencing and Illumina RNA Seq sections in the results. Please rewrite this part.
2.Up to date, four high-quality reference genomes have been assembled in passion fruit. Please refer to the reference genome for gene annotation.
3. The author should select some important PeTPS genes for RT qPCR validation.
4. line 75, the Latin name for yellow passion fruit should be Passiflora edulis f. flavicarpa.

Experimental design

The author should select some important PeTPS genes for RT qPCR validation.

Validity of the findings

NO

Additional comments

NO

Reviewer 2 ·

Basic reporting

The experimental materials were well designed, and the two varieties with flavour differences were selected for full-length transcriptome analysis to obtain better differential expression results; interestingly, the two varieties were completely different after enrichment annotation of up- and down-regulated genes.
(1)The quality of the icon is not high and the clarity is not good.
(2)The reference to aroma in the background section seems to be incomplete and inconsistent with the title and the flavour indicators in the experimental material; flavour includes traits such as sugar-acid and aroma, and should be revised to flavour.
(3)Passifora edulis and P. edulis. are the Latin name, italicised, and the author is requested to correct it in full.
(4)In Figure 1, "b Total acid content (TA, g-kg-1);" appears twice.
(5)The article chose the TPS family for analysis more in consideration of aroma than other flavour-related substances to form related gene families, and the authors are requested to explain.
(6)Figure 9 AtTPS gene should be deleted gene, this figure is made from protein sequences and should be plural.
(7)The genes should be written in italics throughout the text, and the authors are requested to revise the text in its entirety.
(8)Figure 10 b, should be a graph of gene expression trends, not "expression profiles of 17 TPS genes".

Experimental design

(1)The experimental methodology is described in some detail and the raw data are uploaded.

Validity of the findings

(1)The conclusion section is a summary and distillation of the results and should not be a repetition of the results; the authors are requested to make changes.

Additional comments

No.

Reviewer 3 ·

Basic reporting

After reviewing the manuscript, it is evident that the authors have conducted thorough analyses to explore the genetic underpinnings of flavor differences in Passiflora edulis cultivars. The integration of Full-Length Iso-Seq, Illumina RNA-Seq, and flavor testing provides valuable insights into terpene synthase (TPS) gene expression and its association with flavor profiles. However, there are certain areas that require attention.

Experimental design

The Methods section lacks comprehensive coverage and fails to provide sufficient detail and clarity for replication. Essential aspects of the experimental procedures, such as data analysis, and validation techniques, are inadequately described. Check additional comments section

Validity of the findings

no comment

Additional comments

1- The abstract lacks clear enumeration of key results and fails to adequately address the application of the study's findings. It would benefit from succinctly highlighting the main outcomes. Additionally, discussing the practical implications of the results for passion fruit cultivation and flavor enhancement would enhance the abstract's relevance and significance.

2- One notable issue is the lack of detailed explanation regarding terpene synthase (TPS) genes in the introduction section. Given their crucial role on various traits, a more comprehensive overview of TPS genes and their significance in shaping aroma profiles would enhance the reader's understanding of the study's context.

3- Another notable issue observed in the manuscript is inconsistency in italicizing taxonomic names. It is essential to maintain consistency throughout the text to adhere to standard scientific writing conventions (kines 101, 109, 110, 116, 121, etc.)

4- The authors' decision to use Cufflinks for quantification, despite its deprecated status, raises concerns about the reliability and accuracy of the transcript quantification results. Cufflinks, along with its associated tools like Cuffdiff, has indeed been deprecated in favor of newer and more robust RNA-seq analysis tools that offer improved performance and accuracy, such as StringTie and Salmon.Authors should validate the expression of DEGs obtained using Cufflinks by alternative RNA-seq analysis tools for sensitivity analysis

5- Given the complexities involved in identifying lncRNAs and the importance of stringent filtering criteria, I recommend reconsidering the inclusion of the lncRNA identification part in the manuscript. Alternatively, the authors could refer to studies like "Genome-wide screening and characterization of long noncoding RNAs involved in flowering/bolting of Lactuca sativa" and "Identification of the complex interplay between nematode-related lncRNAs and their target genes in Glycine max" for guidance on robust lncRNA identification methodologies. Authors should also prioritize the investigation of lncRNA-target gene interactions to elucidate their regulatory mechanisms and biological significance.

6- Given the importance of confirming TPS genes and their classification, authors should consider further investigation using tools like the TERZYME program and refer to relevant manuscripts such as "Transcriptome Landscape Variation in the Genus Thymus" to enhance the rigor and accuracy of their analysis.

7- The sentence "For phylogenetic analysis, the PeTPSs and AtTPSs obtained were used for classification" lacks clarity regarding the specific methodology employed for phylogenetic analysis. Authors should provide further details on the phylogenetic analysis process, including the software or methods used, alignment procedures, and tree construction techniques.

8- It's crucial for authors to provide a comprehensive overview of the terpene pathway in plants, which includes the identification of all relevant genes involved. The omission of other genes in the terpene pathway may limit the understanding of the pathway's complexity and its relationship to fruit flavor in P. edulis cultivars.

9- Validation of expression results through quantitative real-time PCR (qRT-PCR) is essential to ensure the accuracy and reliability of the findings. Authors should include a section detailing the qRT-PCR validation process, including primer design, experimental setup, and analysis methods.

---

## Round 0.2 · Major Revisions

Thanks for the update and detailed reply to the reviewers. While two reviewers have minor comments, reviewer #3 has some concerns on the methods (Cufflinks and phylogeny). Please use different tools, give the details on the methods.

·

Basic reporting

All my questions have been resolved.

Experimental design

No problem.

Validity of the findings

No problem.

Additional comments

All my questions have been resolved.

Reviewer 2 ·

Basic reporting

The author has made extensive revisions and rationalizations. But there are still some minor issues that need to be revised.
1. The tracked word, Line55-60, change the Passiflora to Passion fruit. Passiflora is a genus name, and in many places throughout the text the genus name is not needed; it would be more appropriate to use passion fruit. The authors are requested to revise the whole text. And the line 164-178 "P. edulis " and so on.
2. Line 772 and 788, The names of the authors of the references should be written in accordance with the requirements of the PeerJ journal.
3. Figure 10 Motif and expression analysis of P. edulis TPS.
P. edulis should be italicized, or PeTPS(italicized) members or PeTPSs(italicized)
4. Figure 9 " AtTPS gene" to "AtTPS members". And " PeTPS gene".

After the authors have made a lot of revisions, please check the above formatting and writing details carefully in the second round of revisions.

Experimental design

no comment

Validity of the findings

no comment

Additional comments

no comment

Reviewer 3 ·

Basic reporting

The manuscript has undergone significant improvements, and the authors have provided valuable insights into the genetic underpinnings of flavor differences in Passiflora edulis cultivars. However, several areas still require further attention.

Experimental design

The Methods section lacks comprehensive coverage and fails to provide sufficient detail and clarity for replication. Essential aspects of the experimental procedures, such as data analysis, and validation techniques, are inadequately described. Check additional comments section

Validity of the findings

no comment

Additional comments

Abstract:
The revised abstract is more comprehensive but still lacks clear enumeration of key results. Additionally, the practical implications of the study's findings for passion fruit cultivation and flavor enhancement should be succinctly highlighted to enhance the abstract's relevance and significance.

Use of Cufflinks for Quantification:
Despite the authors' previous reliance on Cufflinks and Cuffdiff, the continued use of these deprecated tools raises concerns about the reliability and accuracy of the transcript quantification results. The authors should validate the expression of DEGs obtained using Cufflinks with more robust and current RNA-seq analysis tools such as StringTie or Salmon. Using Cufflinks is not acceptable for current standards in RNA-seq analysis.

Classification of Terpene Synthase Genes:
The classification of TPS genes into TPS-a, TPS-b, TPS-c, TPS-e/f, and TPS-g subfamilies requires further rigor. I recommend comparing the data with the TERZYME database to enhance the accuracy of the classification and ensure comprehensive coverage of the TPS gene family.

Phylogenetic Analysis:
The methodology for phylogenetic analysis still lacks some crucial details, such as alignment procedures, quality checks, and trimming methods. A detailed description of these steps, including the specific software and models used, is necessary to ensure the transparency and reproducibility of the analysis.

qRT-PCR Validation:
While the inclusion of qRT-PCR validation is commendable, there is an issue with the reported results. Both RNA-seq and qRT-PCR results are presented according to FPKM values. However, it is unclear how the results of qRT-PCR, typically expressed in Ct values, were converted to FPKM. A clear explanation of the conversion process is necessary to validate the accuracy and reliability of the qRT-PCR results.

---

## Round 0.3 · Minor Revisions

The manuscript has positive comments from the re-reviewers and could be accepted now. But one serious remark from reviewer #1 remains. See: "At present, multiple high-quality reference genomes of passion fruit have been released", and the reference to the databases. Need to compare the data with available reference genomes of passion fruit. It demands some text update, even without re-calculation. Ideally, the revision would contain be re-analysis of sequencing data. Please reply to this comment. We may not send the manuscript for next reviewing round and accept it after the update.

·

Basic reporting

At present, multiple high-quality reference genomes of passion fruit have been released( https://db.cngb.org/search/?q=CNA0017758https://db.cngb.org/search/?q=CNA0017758http://passionfruit.com.cn/cgi-bin/search_genes_expression.pl# )Please align the sequencing data with the reference genome instead of public databases such as NR, Swiss Prot, KOG, Gene Ontology (GO), plantTFDB, Kyoto Encyclopedia of Genes and Genomes (KEGG), and Pfam.

Experimental design

At present, multiple high-quality reference genomes of passion fruit have been released( https://db.cngb.org/search/?q=CNA0017758https://db.cngb.org/search/?q=CNA0017758http://passionfruit.com.cn/cgi-bin/search_genes_expression.pl# )Please align the sequencing data with the reference genome instead of public databases such as NR, Swiss Prot, KOG, Gene Ontology (GO), plantTFDB, Kyoto Encyclopedia of Genes and Genomes (KEGG), and Pfam.

Validity of the findings

no.

Additional comments

no.

Reviewer 2 ·

Basic reporting

no comment

Experimental design

no comment

Validity of the findings

no comment

Additional comments

Accept

Reviewer 3 ·

Basic reporting

no comment

Experimental design

no comment

Validity of the findings

no comment

Additional comments

no comment

---

## Round 0.4 · accepted · Accept

Thanks for the manuscript update and detailed answer.